# PROTEIN SEQUENCE DOMAIN ANNOTATION USING LANGUAGE MODELS

## ABSTRACT

Protein function inference relies on annotating protein domains via sequence similarity, often modeled through profile Hidden Markov Models (profile HMMs), which capture evolutionary diversity within related domains. However, profile HMMs make strong simplifying independence assumptions when modeling residues in a sequence. Here, we introduce PSALM (Protein Sequence Annotation using Language Models), a hierarchical approach that relaxes these assumptions and uses representations of protein sequences learned by protein language models to enable high-sensitivity, high-specificity residue-level protein sequence annotation. We also develop the Multi-Domain Protein Homology Benchmark (MDPH-Bench), a benchmark for protein sequence domain annotation, where training and test sequences have been rigorously split to share no similarity between any of their domains at a given threshold of sequence identity. Prior benchmarks, which split one domain family at a time, do not support methods for annotating multi-domain proteins, where training and test sequences need to have multiple domains from different families. We validate PSALM's performance on MDPH-Bench and highlight PSALM as a promising alternative to HMMER, a state-of-the-art profile HMM-based method, for protein sequence annotation.

## 1 INTRODUCTION

Proteins are composed of distinct structural and functional units conserved through evolution, known as domains. The primary aim of protein sequence annotation is to locate and characterize these domains within a given sequence. Insight into the individual functions of these domains, which may act independently or in concert with neighboring domains, may shed light on the overall biological role of the protein (Fig. 1). Since experimental characterization of protein function can be difficult, function is often inferred and annotated through sequence similarity (homology) to domains with known function (Pearson, 2013; Eddy, 1998). As the size of protein sequence databases and the number of protein sequences with unknown function continue to grow rapidly (UniProt Consortium, 2023), methods for annotating protein sequences have been critical for exploiting this wealth of information about the molecular basis and evolutionary trajectory of life.

The state of the art in protein domain sequence annotation uses profile hidden Markov models (profile HMMs) to detect domains (Eddy, 2011) and profile/profile comparison to identify homologous domains (Remmert et al., 2012). Databases of protein domain families, like Pfam (Mistry et al., 2021), categorize millions of protein sequences into approximately 20,000 domains. The state of the art uses profile hidden Markov models (profile HMMs) to identify domains (Eddy, 2011) and profile/profile comparison to identify homologous domains (Remmert et al., 2012). Annotation can be achieved by using databases of protein families to compare a protein sequence against 20,000 profiles rather than scanning against millions of individual sequences. Domain-based annotation goes beyond classifying the whole protein sequence; it identifies the domain composition as well as the boundaries of each domain. This "domain annotation" requires annotation at the residue level, labeling each amino acid symbol in the sequence. Domain annotation helps with function inference and avoids the "transitive identification catastrophe", where sequence-level annotations inferred from the presence of one domain can erroneously transfer between sequences due to homology of an unrelated domain (Doerks et al., 1998).

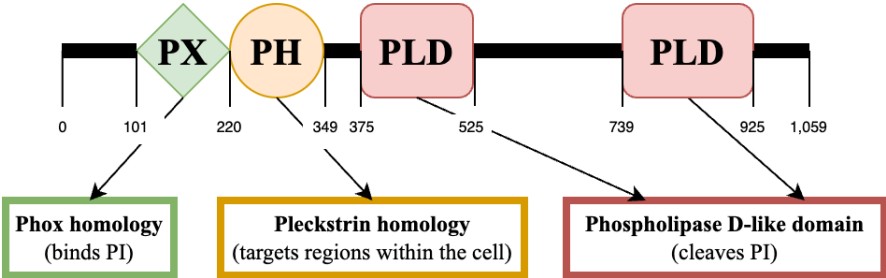

Figure 1: **Annotated domain architecture of a human phospholipase D1 protein** (Q59EA4) (EMBL-EBI, 2024; Paysan-Lafosse et al., 2023), featuring PX (phox), PH (pleckstrin homology), and PLD (phospholipase D-like) domains. Together, the function of these domains suggest that the full length protein (1,059 amino acids) is involved in phosphatidylcholine (PI) cleavage and intracellular signaling, consistent with experimental evidence.

Profile HMMs make simplifying independence assumptions when modeling residues in protein sequences, and there is considerable interest in developing more powerful methods that can better recognize distant evolutionary relationships with greater sensitivity. While most deep learning approaches to protein sequence similarity recognition focus on whole-protein or single-domain classification (Bileschi et al., 2022; Heinzinger et al., 2022; Nallapareddy et al., 2023; Kaminski et al., 2023; Hamamsy et al., 2023), they do not address the challenge of identifying individual domain subsequences within longer target sequences, which requires careful benchmarking and data curation to assess performance.

In this work, we introduce Protein Sequence Annotation with Language Models (PSALM), a novel approach that extends the capabilities of ESM-2, a pre-trained protein language model (pLM) (Lin et al., 2023), to predict *residue-level* sequence annotations. Our contributions include:

- **First deep learning model for residue-level protein domain annotation:** PSALM is the first deep learning approach to annotate domain boundaries and subsequences within multidomain proteins.
- **Relaxation of HMM independence assumptions for improved sensitivity and specificity:** PSALM leverages pLMs to overcome the simplifying assumptions of profile HMMs, allowing for greater sensitivity in detecting conserved domains across distantly-related sequences and higher specificity in identifying previously unannotated domains.
- **First benchmark for multidomain protein annotation, MDPH-Bench:** To enable robust evaluation, we introduce the Multi-Domain Protein Homology Benchmark (MDPH-Bench). This benchmark rigorously curates training and test sets to prevent any domain similarity above a predefined threshold, enabling realistic assessments of model performance across diverse domain families and multidomain proteins, which previous benchmarks do not support.

## 2 RELATED WORK

### 2.1 PROFILE HMMS

Profile HMMs use curated multiple sequence alignments (MSAs) of related domains, which reveal patterns of conservation and variability at the residue level, to model consensus using "match", "insert", and "delete" hidden states (Durbin et al., 1998; Eddy, 1998). Profile HMMs assume that the observed residues are conditionally independent given the hidden state. While this assumption simplifies the modeling process, it may limit the ability of profile HMMs to capture complex dependencies between residues in a sequence. These models serve as templates for comparison against the sequence of interest, enabling the identification of domains by finding subsequences that match the profile HMMs. Sequences with multiple, unrelated domains will require the use of multiple profile HMMs for annotation. HMMER (Eddy, 2011) is the state-of-the-art protein sequence domain annotation method and underlies many different databases, which organize related domains

into MSAs and profile HMMs at varying levels of granularity, enabling profile-based annotations at the superfamily (Pandurangan et al., 2019), family (Mistry et al., 2021), and sub-family (Thomas et al., 2022) levels.

## 2.2 Deep Models

Recent efforts to apply deep learning methods to predict protein function from sequence have either focused on predicting ontology-based functional annotation at the sequence level (Cao & Shen, 2021; Hong et al., 2020; Kulmanov & Hoehndorf, 2020) or recognizing homology at the sequence level (Heinzinger et al., 2022; Nallapareddy et al., 2023; Kaminski et al., 2023; Hamamsy et al., 2023). To our knowledge, ProtENN, an ensemble of convolutional neural networks, represents the first attempt to predict Pfam domains directly from protein sequences (Bileschi et al., 2022). ProtENN, however, is constrained to make one domain prediction per input sequence and cannot natively identify domain boundaries or multiple domains within a sequence without ad hoc post-processing. Additionally, ProtENN cannot provide information on the contribution of an individual residue to a predicted annotation.

## 3 Methods

### 3.1 Problem Formulation

Here, we formalize the residue-level sequence annotation problem as a mapping from a protein sequence $\mathbf{x} = (x_1, x_2, \ldots, x_L)$ to a sequence of protein domain families $\mathbf{y} = (y_1, y_2, \ldots, y_L)$. For residue $i$ in a sequence, $x_i$ is an index $1 \ldots 25$ representing the $i$-th amino acid character (20 canonical, 2 non-canonical, and 3 ambiguous amino acid characters), and $y_i$ is an index $1 \ldots D$ representing the $i$-th protein domain family annotation, with $D + 1$ for none. Approximately 23% of protein sequences and 47% of residues across all sequences in UniProt do not belong to any Pfam domain family (Mistry et al., 2021). The goal of residue-level annotation is to learn a model that predicts domain family annotations for each residue in a protein sequence:

$$\hat{y}_i = \arg\max_f P(Y_i = f|\mathbf{x}), \tag{1}$$

where $P(Y_i)$ is the distribution over $D + 1$ family annotations for a given residue $i$.

### 3.2 Protein Language Model

Numerically encoding protein sequences is necessary as an input for machine learning tasks such as classification. Protein language models (pLMs) learn vector representations of both individual residues and full-length protein sequences, which capture long-range interactions, predict function via transfer learning, and achieve state-of-the-art performance in several structure prediction tasks Bepler & Berger (2021); Rao et al. (2020); Meier et al. (2021); Elnaggar et al. (2021). We use ESM-2 (specifically, the 8M, 35M, 150M, and 650M parameter models), a pre-trained general-purpose pLM (Lin et al., 2023), to generate residue-level sequence embeddings $\mathbf{x}'$ for a given sequence $\mathbf{x}$. ESM-2 was trained using a BERT-style masked token prediction task (Devlin et al., 2018), enabling it to capture contextual information and dependencies within protein sequences and allowing us to replace $\mathbf{x}$ with $\mathbf{x}'$.

### 3.3 PSALM

We introduce PSALM (Protein Sequence Annotation using Language Models), a method to predict domains across a protein sequence at the residue-level. PSALM uses a hierarchical approach that considers both individual protein domain families and clans, which are collections of evolutionarily related (homologous) protein domain families categorized by Pfam (Finn et al., 2006). In Pfam 35.0, approximately 45% of the 19,632 Pfam families are grouped into 655 clans, and a family can only belong to at most one clan. While our primary aim is to predict protein domain families at each residue, modeling clans – the super class – is an interpretable intermediate step that aids in identifying areas of functional or structural importance that may not have clear family-level annotations.

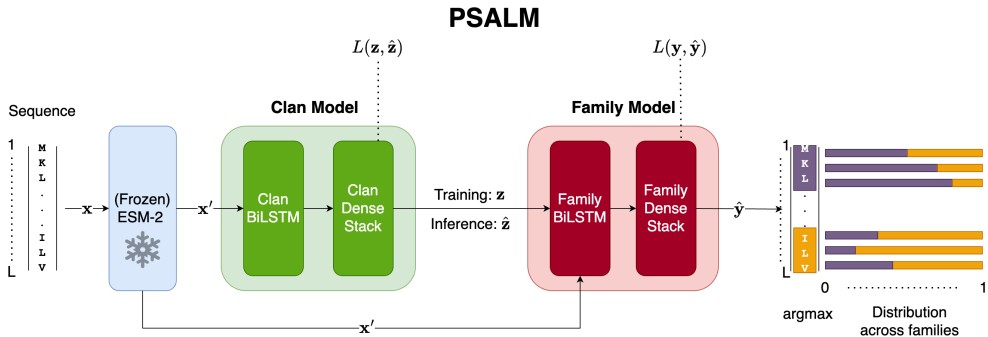

Figure 2: **Overview of residue-level protein sequence annotation with PSALM.** A sequence $\mathbf{x}$ of length $L$ is embedded as $\mathbf{x}'$ with a frozen ESM-2. The PSALM clan and family models predict the clan annotations $\hat{\mathbf{z}}$ and family annotations $\hat{\mathbf{y}}$, respectively, and are trained to minimize cross-entropy loss $L(\cdot)$. Here, the example outputs are predicted across a set of 2 families.

This intermediate annotation problem is a mapping from $\mathbf{x}$ to a sequence of Pfam clans $\mathbf{z} = (z_1, z_2, \ldots, z_L)$, where $z_i$ is an index $1 \ldots C$ representing the $i$-th clan annotation, with $C + 1$ for "non-clan" or $C + 2$ for none. The non-clan annotation describes a residue which belongs to a domain family that is not a member of clan, and none refers to a residue which does not belong to a domain family and thus does not belong to a clan. For a given residue, the PSALM clan and family models learn to predict:

$$\hat{z}_i = \arg\max_c P(Z_i = c | \mathbf{x}') \tag{2}$$

$$\hat{y}_i = \arg\max_f P(Y_i = f | Z_i = C(f), \mathbf{x}') P(Z_i = C(f) | \mathbf{x}'), \tag{3}$$

where $C(f)$ is the clan label to which family $f$ belongs, and $P(Z_i)$ is the distribution over all $C + 2$ clan annotations for a given residue $i$. The inclusion of a separate clan prediction task ensures the interpretability of the clan model, preventing it from becoming an abstract hidden state. The family model is trained via the teacher forcing algorithm (Williams & Zipser, 1989), where it is provided the correct clan annotation for each residue in order to mitigate error propagation.

The clan and family models follow a similar structure and are trained separately. Protein sequences are initially embedded at the residue level using a pre-trained and frozen instance of ESM-2. The resulting embeddings are then passed into a bidirectional Long Short-Term Memory (BiLSTM) layer to capture sequential dependencies (Hochreiter & Schmidhuber, 1997) in the forwards and backwards directions, each of which, like profile HMMs, are strongly linear, left-right models (Eddy, 1998). The choice of BiLSTM was made deliberately to introduce as few changes as possible, ensuring that the observed performance improvements could be attributed primarily to the use of the protein language model, rather than architectural differences from profile HMMs. The output from the BiLSTM layer is subsequently decoded using a stack of three dense layers, scaled to the number of clans or family labels, to produce logits across the prediction space. Probabilities are computed by applying softmax to the logits generated by each model.

## 4 BENCHMARK

In many machine learning contexts, data samples are often assumed to be independent instances drawn from a distribution of the data, justifying random training-test splits. However, this assumption does not hold for sequences in protein domain families, which share evolutionary relationships. Random data splits for protein sequences may lead to performance overestimation, motivating the need to explicitly consider sequence similarity when partitioning data into distinct training and test sets (Söding & Remmert, 2011; Walsh et al., 2016; Jones, 2019; Walsh et al., 2021; Petti & Eddy, 2022). For MDPH-Bench, we aim to create a benchmark that simulates the challenges posed by

Table 1: MDPH-Bench test subset, training, and validation details

| Test splits | 0-20% | 20-40% | 40-60% | 60-80% | 80-100% | Train | Val |
|---|---|---|---|---|---|---|---|
| Sequences | 4,087 | 37,319 | 17,446 | 8,570 | 5,864 | 517,936 | 5,775 |
| Families | 543 | 2,456 | 1,952 | 1,731 | 1,697 | 14,811 | 2,097 |
| Clans | 180 | 414 | 365 | 341 | 319 | 646 | 388 |
| Coverage | 65.31% | 59.74% | 58.66% | 62.02% | 56.71% | 60.41% | 58.79% |
| Average PID | 18.35% | 27.64% | 42.45% | 58.08% | 80.01% | NA | 45.08% |

the remote homology detection problem, where previously unknown or unannotated sequences with little similarity to the training set are especially difficult to detect and annotate. To address this, the MDPH-Bench test set includes a diverse selection of multi-domain proteins, spanning a wide range of sequence similarity to the training set.

## 4.1 BENCHMARK CREATION

We begin by collecting the 1.2M "seed domains" from Pfam-A Seed 35.0, a set of curated, representative domains for each domain family that are used to build the 20K Pfam profile HMMs (Mistry et al., 2021). We apply BLUE (Petti & Eddy, 2022), a graph-based sequence splitting algorithm, to partition the seed domains into preliminary training and test sets, defining an edge between two domains as their pairwise percent identity (Appendix A.1.1). We use a PID threshold of 25% to split the seed domains, resulting in 560K training domains and 190K test domains. The remaining 450K seed domains were discarded due to sharing $> 25\%$ PID with both test and training sets. We then retrieve the full-length sequences corresponding to these representative training and test domains from UniProt release March 2021, a comprehensive database of 230M protein sequences. This results in 517K training sequences. From the test set, we eliminate duplicate sequences also present in the training set. All sequences across both training and test sets are annotated via the `hmmscan` tool from HMMER (Eddy, 2011) with strict inclusion thresholds (E-value < 0.001, bitscore $\geq$ 30) in order to identify domain hits that constitute a "ground truth", with special care to nested, contiguous domains, which may escape typical processing methods (Appendix A.1.2). For training and test purposes, family and clan labels are only assigned to ground truth domains. We discard sequences in test that do not contain an annotated ground truth domain represented by at least 20 ground truth domains from the same family in the training set, resulting in 73K test sequences. For each test sequence, we compute the maximum PID between any of its domains to any domain in the training set as a proxy for its distance from the training set (Appendix A.1.3). We partition the test set into five subsets based on this maximum PID (Table 1), and a total of 6K validation sequences are sampled uniformly across the test subsets. Such a partition may result in test sequences that, for example, may be placed in the $80 < \text{PID} \leq 100$ subset due to a single domain closely related to one in the training set, whereas the test sequence may have several other domains that share significantly lower PID with domains in the training set (this is why the average PID is near the lower bound of the max PID range for many of the test subsets in Table 1). The domain coverage, defined as the average percent of residues in a sequence that are labeled by Pfam domains, is similar across all test subsets.

## 4.2 ADDRESSING POSSIBLE LEAKAGE

We address the potential for unannotated domains to introduce data leakage across the training and test sets by shuffling all subsequences without family and clan labels in the test sequences to disrupt possible domain structures, preserving $0^{th}$ order residue-composition (Pearson, 2013; Eddy, 2011). The ground truth for protein sequence annotation is fundamentally unknown, relying on inference rather than complete structural and evolutionary knowledge – this is why we must assume natrual sequences contain unannotated true domains that new methods may discover. Since PSALM may be sensitive enough to identify unannotated domains, it is trained with these regions shuffled, to mitigate penalties for "false positives" (with respect to the ground truth annotations). Another source of data leakage may arise from the millions of representative sequences from the UniRef50 database release April 2021 (Suzek et al., 2015) used to train ESM-2. We identify that none of the 4,087 sequences in the $0 < \text{PID} \leq 20$ test subset were present as representative sequences in this version of UniRef50. However, UniRef50 may contain close homologs to the sequences in this test subset.

| Model | # Params | | Learning Rate | |
| --- | --- | --- | --- | --- |
| | Clan | Family | Clan | Family |
| $PSALM_{650}$ | 69M | 166M | $5e-4$ | $5e-5$ |
| $PSALM_{150}$ | 18M | 67M | $5e-4$ | $5e-5$ |
| $PSALM_{35}$ | 10M | 47M | $5e-4$ | $5e-5$ |
| $PSALM_8$ | 5M | 29M | $5e-4$ | $5e-5$ |
| $PSALM_{OH}$ | 56M | 153M | $1e-4$ | $1e-5$ |

Table 2: Number of parameters and learning rates (LR) for PSALM models.

## 5 RESULTS

### 5.1 BASELINES

We establish two baseline methods for comparison. We use HMMER, the current state-of-the-art protein sequence annotation method, to build profile HMMs from MSAs of the ground truth domains in the training set, denoted as HMMER*, and use these profiles to annotate the test sequences with `hmmscan`. This allows us to evaluate how a state-of-the-art profile HMM method compares to PSALM when using the same training and testing sets. Additionally, we implement a variant of PSALM, denoted as $PSALM_{OH}$, where one-hot embeddings for each amino acid in a protein sequence are utilized instead of embeddings from the pre-trained protein language model ESM-2. This comparison helps discern whether differences in performance between PSALM and HMMER* are influenced by ESM-2 or solely by the subsequent neural network architecture. Additionally, we assess the scalability of PSALM by evaluating performance across different ESM-2 model sizes (8M, 35M, 150M, and 650M parameters).

More recent deep learning approaches (e.g., ProtENN) are not included as baselines because they address sequence-level or single-domain classification, which is a different problem that is largely irrelevant to domain- or residue-level annotation. We discuss the difference between the two in our related work section (Section 2.2). Biologists prefer domain-level annotation for many reasons, including avoiding the "transitive catastrophe" (lines 27-33), where unrelated sequences cluster through homologous domains (e.g., protein AB shares homology with BC, BC with CD, and all three cluster; but AB shares no homology with CD). Biologists rely on extensive domain-level annotation resources built on a state of the art of profile HMMs.

### 5.2 IMPLEMENTATION DETAILS

Both PSALM and PSALM-onehot are trained using cross entropy loss over the entire sequence for both family and clan annotations. For training all PSALM+ESM-2 models, we use ADAM optimizer (Kingma & Ba, 2014) with initial learning rate $5e-4$ for the clan model and $5e-5$ for the family model. These values were selected via hyperparameter tuning from across the following learning rates: $[1e-3, 5e-4, 1e-4, 5e-5, 1e-5]$. A similar hyperparameter search results in a learning rate of $1e-4$ for the $PSALM_{OH}$ clan model and $1e-5$ for the family model. We employ a learning rate scheduler that reduces the learning rate by a factor of $\sqrt{10}$ if the validation loss fails to decrease over consecutive epochs with an additional early stopping criterion of 5 epochs with no improvement. The effective batch size is 32,768 tokens.

The number of parameters for all PSALM clan and family models are given in Table 2. All models were trained on four NVIDIA A100 80GB GPUs. To accommodate memory limitations on the GPU, all sequences are truncated to a maximum length of 4096 residues. This truncation strategy only applies to approximately 0.25% of sequences across the training and test sets and does not reflect a model limitation – PSALM can be used to annotate sequences of any length provided enough memory. All procedures from the HMMER tool suite use version $3.4$ (Eddy, 2011).

### 5.3 METRICS

Protein sequence databases have vastly more negatives than positives, requiring extremely low (essentially zero) and controllable false positive rate (FPR), as false annotations are amplified and

Table 3: PSALM MDPH-Bench residue-level domain annotation results

| PID | Model | Clan | | | | Family | | | |
|-----|-------|------|-----|-----|------|--------|-----|-----|------|
| | | TPR | FPR | F1 | MCC | TPR | FPR | F1 | MCC |
| 0-20% | HMMER* | 0.694 | 0.033 | 0.819 | 0.642 | 0.659 | 0.033 | 0.810 | 0.636 |
| | $PSALM_{650}$ | **0.944** | **0.022** | **0.985** | **0.957** | **0.750** | **0.012** | **0.978** | **0.947** |
| | $PSALM_{150}$ | 0.862 | 0.133 | 0.912 | 0.758 | 0.621 | 0.050 | 0.869 | 0.730 |
| | $PSALM_{35}$ | 0.729 | 0.174 | 0.847 | 0.620 | 0.428 | 0.071 | 0.721 | 0.532 |
| | $PSALM_8$ | 0.589 | 0.214 | 0.772 | 0.488 | 0.211 | 0.079 | 0.463 | 0.293 |
| | $PSALM_{OH}$ | 0.490 | 0.100 | 0.764 | 0.559 | 0.089 | 0.022 | 0.236 | 0.203 |
| 20-40% | HMMER* | 0.907 | 0.043 | 0.941 | 0.862 | **0.876** | 0.043 | 0.939 | 0.861 |
| | $PSALM_{650}$ | **0.966** | **0.020** | **0.985** | **0.964** | 0.845 | **0.015** | **0.982** | **0.959** |
| | $PSALM_{150}$ | 0.887 | 0.092 | 0.925 | 0.819 | 0.727 | 0.036 | 0.910 | 0.817 |
| | $PSALM_{35}$ | 0.799 | 0.131 | 0.873 | 0.709 | 0.607 | 0.056 | 0.825 | 0.682 |
| | $PSALM_8$ | 0.636 | 0.192 | 0.788 | 0.553 | 0.353 | 0.074 | 0.623 | 0.452 |
| | $PSALM_{OH}$ | 0.516 | 0.107 | 0.780 | 0.602 | 0.102 | 0.023 | 0.282 | 0.251 |
| 40-60% | HMMER* | 0.951 | 0.058 | 0.957 | 0.898 | 0.921 | 0.058 | 0.956 | 0.896 |
| | $PSALM_{650}$ | **0.977** | **0.020** | **0.986** | **0.966** | **0.924** | **0.017** | **0.984** | **0.964** |
| | $PSALM_{150}$ | 0.888 | 0.058 | 0.927 | 0.834 | 0.806 | 0.026 | 0.919 | 0.832 |
| | $PSALM_{35}$ | 0.826 | 0.101 | 0.882 | 0.736 | 0.728 | 0.049 | 0.866 | 0.738 |
| | $PSALM_8$ | 0.704 | 0.158 | 0.809 | 0.598 | 0.532 | 0.072 | 0.741 | 0.567 |
| | $PSALM_{OH}$ | 0.666 | 0.104 | 0.835 | 0.671 | 0.159 | 0.029 | 0.430 | 0.363 |
| 60-80% | HMMER* | 0.974 | 0.059 | 0.971 | 0.924 | 0.946 | 0.059 | 0.970 | 0.923 |
| | $PSALM_{650}$ | **0.984** | **0.018** | **0.988** | **0.970** | **0.957** | **0.016** | **0.988** | **0.968** |
| | $PSALM_{150}$ | 0.912 | 0.058 | 0.940 | 0.850 | 0.859 | 0.028 | 0.936 | 0.852 |
| | $PSALM_{35}$ | 0.845 | 0.083 | 0.900 | 0.761 | 0.782 | 0.045 | 0.888 | 0.759 |
| | $PSALM_8$ | 0.728 | 0.154 | 0.827 | 0.609 | 0.605 | 0.084 | 0.778 | 0.583 |
| | $PSALM_{OH}$ | 0.788 | 0.094 | 0.890 | 0.745 | 0.216 | 0.027 | 0.573 | 0.478 |
| 80-100% | HMMER* | 0.977 | 0.051 | 0.972 | 0.935 | 0.950 | 0.051 | 0.971 | 0.934 |
| | $PSALM_{650}$ | **0.981** | **0.015** | **0.986** | **0.969** | **0.967** | **0.012** | **0.986** | **0.968** |
| | $PSALM_{150}$ | 0.895 | 0.049 | 0.929 | 0.845 | 0.851 | 0.024 | 0.924 | 0.845 |
| | $PSALM_{35}$ | 0.812 | 0.088 | 0.872 | 0.732 | 0.732 | 0.046 | 0.853 | 0.725 |
| | $PSALM_8$ | 0.711 | 0.114 | 0.809 | 0.624 | 0.601 | 0.059 | 0.761 | 0.600 |
| | $PSALM_{OH}$ | 0.877 | 0.066 | 0.925 | 0.836 | 0.282 | 0.018 | 0.709 | 0.630 |

propagated to additional sequences by later searches. Methods in this field are typically benchmarked for the sensitivity or true positive rate (TPR) they can achieve at a high specificity (low FPR). We also report the F1 score and Matthews Correlation Coefficient (MCC). Here, FPR is defined as the fraction of true negative residues (shuffled, preserving residue composition) incorrectly identified as homologous to a Pfam protein domain family, and TPR is defined as the fraction of residues in ground truth domains correctly identified.

## 5.4 EVALUATION

We highlight the key observations from the sequence annotation benchmark (Table 3). $PSALM_{650}$ demonstrates superior performance in residue-level domain annotation, accurately annotating a substantial portion of true domain regions while consistently calling fewer false positives compared to HMMER*. Specifically, $PSALM_{650}$ reaches higher TPR, F1 and MCC scores at a lower FPR than HMMER*, with the single exception being family TPR at the 20-40% max PID range test subset. The performance of $PSALM_{650}$ is especially noteworthy in the 0-20% max PID range test subset, which constitutes the most difficult to detect domains in MDPH-Bench, as these sequences share very little max sequence similarity with any domain in the training set – $PSALM_{650}$ is much more sensitive and specific than HMMER*. $PSALM_{OH}$, the baseline to ablate the significance of the ESM-2 contextual residue-level embeddings, learns clan-level residue annotations comparably to $PSALM_8$ and $PSALM_{35}$, especially at the higher max PID range test subsets, but it performs relatively poorly at the family level though it has a consistently low FPR. Additionally, the scaling

Table 4: Family-only PSALM MDPH-Bench results at a fixed FPR of 0.01

| PID | Model | Clan | | | | Family | | | |
|---|---|---|---|---|---|---|---|---|---|
| | | TPR | F1 | MCC | AUC | TPR | F1 | MCC | AUC |
| 0-20% | HMMER* | 0.662 | 0.796 | 0.629 | 0.681 | 0.636 | 0.790 | 0.625 | 0.649 |
| | $\text{PSALM}_{650}$ | **0.922** | **0.970** | **0.920** | **0.934** | **0.735** | **0.942** | **0.868** | **0.744** |
| | $\text{PSALM\_F}_{650}$ | 0.692 | 0.821 | 0.659 | 0.697 | 0.628 | 0.806 | 0.649 | 0.630 |
| 20-40% | HMMER* | 0.797 | 0.885 | 0.774 | 0.896 | 0.774 | 0.881 | 0.771 | **0.867** |
| | $\text{PSALM}_{650}$ | **0.941** | **0.970** | **0.931** | **0.955** | **0.811** | **0.934** | **0.862** | 0.830 |
| | $\text{PSALM\_F}_{650}$ | 0.779 | 0.877 | 0.764 | 0.780 | 0.746 | 0.873 | 0.760 | 0.746 |
| 40-60% | HMMER* | 0.502 | 0.666 | 0.530 | 0.882 | 0.494 | 0.662 | 0.527 | 0.857 |
| | $\text{PSALM}_{650}$ | **0.953** | **0.974** | **0.939** | **0.968** | **0.889** | **0.951** | **0.894** | **0.909** |
| | $\text{PSALM\_F}_{650}$ | 0.827 | 0.903 | 0.807 | 0.831 | 0.812 | 0.901 | 0.806 | 0.815 |
| 60-80% | HMMER* | 0.487 | 0.653 | 0.503 | 0.885 | 0.474 | 0.646 | 0.499 | 0.861 |
| | $\text{PSALM}_{650}$ | **0.967** | **0.981** | **0.952** | **0.977** | **0.934** | **0.970** | **0.927** | **0.947** |
| | $\text{PSALM\_F}_{650}$ | 0.883 | 0.936 | 0.854 | 0.886 | 0.872 | 0.935 | 0.853 | 0.874 |
| 80-100% | HMMER* | 0.525 | 0.687 | 0.558 | 0.891 | 0.515 | 0.683 | 0.555 | 0.866 |
| | $\text{PSALM}_{650}$ | **0.972** | **0.983** | **0.962** | **0.977** | **0.961** | **0.981** | **0.958** | **0.964** |
| | $\text{PSALM\_F}_{650}$ | 0.892 | 0.940 | 0.875 | 0.892 | 0.887 | 0.939 | 0.875 | 0.887 |

analysis across the different ESM-2 model sizes (and PSALM sizes, which scale to the selected ESM-2 model) demonstrates increased performance as model size increases up to $\text{PSALM}_{650}$.

We conduct an additional ablation experiment in order to study the effect of predicting clan-level annotations as an interpretable intermediate in PSALM (Table 4). We retrain the $\text{PSALM}_{650}$ family model without providing any predicted clan annotations and denote this model as $\text{PSALM\_F}_{650}$. We compare $\text{PSALM\_F}_{650}$ to HMMER* and $\text{PSALM}_{650}$ at a fixed residue-level FPR of 0.01 (the full family-only PSALM table is given in Appendix A.2). We additionally report normalized AUC at this fixed FPR. It is not possible to report AUC without fixing the FPR across all methods, since it is not feasible for the residue-level FPR to reach 1 for HMMER(*) and PSALM when no alignment or an explicit "none" class is predicted, respectively. Clan predictions are generated by identifying the clan corresponding to the predicted family. $\text{PSALM}_{650}$ outperforms HMMER* and $\text{PSALM\_F}_{650}$ in almost all metrics across all test subsets, demonstrating the advantage of predicting clan annotations prior to family annotations. The clan-level and family-level performance of $\text{PSALM\_F}_{650}$ are very similar for a given test subset, which is not the case for $\text{PSALM}_{650}$, where the clan-level performance is much higher than the family-level performance, especially at the low max PID range test subsets, which represent sequences more distantly related to the training set. This implies that the erroneous family annotations that $\text{PSALM\_F}_{650}$ predicts are not members of the correct clan, which is typically not the case for $\text{PSALM}_{650}$.

## 6 EXAMPLES

In Figure 3, we compare PSALM annotations with those from InterPro (Paysan-Lafosse et al., 2023), a database integrating domain annotations from various sources, focusing on two protein sequences drawn from the test set. This comparison provides insights into PSALM's performance, highlighting its its ability to annotate multiple domains and its high sensitivity in detecting domains.

PSALM is able to identify multiple domains within a single sequence and accurately annotates the three domains in the multi-domain G2/mitotic-specific cyclin-A protein (Fig. 3 A). While most predicted domain boundaries closely match the database annotations, including the extremely short three amino acid region between the N-and C-terminal cyclin domains, some residues that extend past the edges of the first domain (Cyclin-A N-terminal APC/C binding region) exhibit lower assignment probability. This reflects uncertainty in boundary delineation, a common occurrence across annotation methods, which typically differ by a few residues when identifying exact domain boundaries.

PSALM is highly-sensitive and is able to annotate domains that Pfam profile HMMs miss but other databases (and their models) hit as well as domains missed by all methods, as illustrated by the

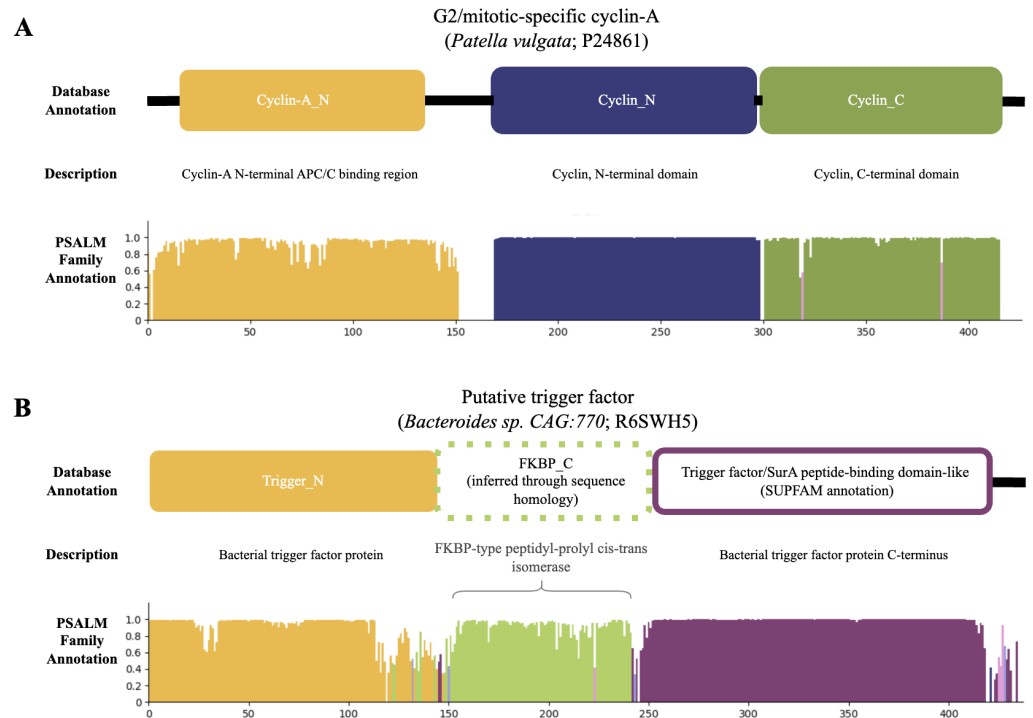

Figure 3: **Comparison of PSALM annotations with those from InterPro for three selected protein sequences. A)** PSALM family annotations on a G2/mitotic-specific cyclin-A protein (Patella vulgata, P24861) from the 80-100% max PID test subset. **B)** PSALM family annotations on a putative trigger factor protein (Bacteroides sp. CAG:770, R6SWH5) from the 20-40% max PID test subset. Filled domains represent InterPro annotations corresponding to Pfam domains. The solid boundary shows a domain from SUPFAM, and the dashed boundary represents a homology-inferred domain not found in InterPro. Approximately 30,000 InterPro proteins share the predicted domain architecture, including trigger factor protein P44837 (Haemophilus influenzae).

putative trigger factor protein example (Fig. 3 B). PSALM identifies both N- and C-terminal bacterial trigger factor domains. This is consistent with the integrated InterPro annotation, even though Pfam lacks a C-terminal annotation for this protein.[1] Additionally, PSALM annotates a middle domain that correspond to a FKBP-type isomerase domain, which is missed by all methods for this sequence – we validate this annotation by identifying 30K proteins in InterPro that share this domain architecture (exact order of these three domains) and sequence length.

## 7 CONCLUSIONS

We introduce PSALM, a highly sensitive and specific pLM-based protein sequence annotation method. PSALM extends the capabilities of self-supervised pLMs with just a few hundred thousand protein sequences, enabling interpretable residue-level annotations at both the clan and family levels. Comparisons with InterPro show PSALM's ability to detect multiple domains, including those currently unannotated. Ablation experiments confirm the importance of pLM embeddings over one-hot encodings and the importance of clan annotations in achieving higher family-level sensitivity and specificity. We find that PSALM performance improves with larger ESM-2 models. We also introduce MDPH-Bench, the first protein sequence benchmark that splits training and test sequences by domain-level sequence similarity for multi-domain proteins. This benchmark minimizes data leakage and enables model evaluation across an evolutionarily-diverse set of proteins.

---

[1]Putative trigger factor protein R6SWH5 is no longer available in UniProt as of November 7th, 2023. However, the entry and all annotations can be accessed either through UniParc (UniProt Consortium, 2023) or by reannotating with InterProScan (Paysan-Lafosse et al., 2023).

Model training and evaluation code as well as sequence IDs for MDPH-Bench are provided in the supplementary material. Model weights and full sequences are too large to be included in the supplementary material at the time of submission. After the review period, all code, data, and weights will be made available online to support reproducibility and protein sequence domain annotation.

# 8    LIMITATIONS & FUTURE WORK

We address the current limitations and potential future research directions of this approach with the following points.

## 8.1    DATA LEAKAGE

Despite our efforts to mitigate it, information from the test set may still contribute to training through the millions of sequences used to train ESM-2. While we exclude sequences used to train ESM-2 from our test subset with the lowest maximum PID, homology could still lead to indirect leakage. However, our maximum PID guarantees help minimize this risk. Ideally, retraining ESM-2 on our training data would further alleviate this concern. We have not done this in the present work because of the compute demand for training a pLM like ESM-2, but we hope to rigorously split a much larger set of proteins to train a pLM from scratch.

## 8.2    DOMAIN CALLING

PSALM cannot distinguish between repeated domains occurring consecutively or accurately resolve split domains. For example, if a domain repeats immediately after itself, PSALM labels the entire two domain block instead of recognizing two separate domains within it. Similarly, when a domain is split, PSALM identifies the two halves as separate domains from the same family, rather than as originating from a single domain. We aim to address this by explicitly modeling the domain boundaries and developing domain-calling algorithms.

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

# A APPENDIX

## A.1 BENCHMARK CREATION

### A.1.1 BLUE

We use the BLUE algorithm (Petti & Eddy, 2022) to split the 1.2M Pfam Seed domains into preliminary train and test sets with a PID threshold of 25%. The pairwise PID between two sequences $\mathbf{x}$ and $\mathbf{y}$ is defined as follows:

$$\text{PID}(\mathbf{x}, \mathbf{y}) = \frac{\text{\# aligned residues}}{\min(\ell(\mathbf{x}), \ell(\mathbf{y}))}, \tag{4}$$

where $\ell(\mathbf{x})$ and $\ell(\mathbf{y})$ represent the lengths of sequences $\mathbf{x}$ and $\mathbf{y}$, respectively. If two domains are in the same family, PID is directly calculated from their seed alignment. If two domains are not in the same family but are in the same clan, they are aligned using the `glsearch` tool from the FASTA3 software package (Pearson, 1999), which performs a "global-local" alignment to account for possible large differences in sequence length. If two domains are not in the same clan, they are assumed to share $< 25\%$ PID.

### A.1.2 GROUND TRUTH ANNOTATION

We determine "ground truth" by annotating full length sequence with Pfam Seed profile HMMs using `hmmscan` with strict inclusion criteria (E-value $< 0.001$, bitscore $\geq 30$). The highest-scoring annotation at each residue is taken as ground truth, but additional post-processing is necessary to ensure that "nested" domain structures are retained. This is accomplished by considering the "match strings" that HMMER generates for an alignment. The match strings contain characters that represent matches, where residues align to a given domain profile, and characters that represent inserts, where unaligned residues are inserted into the sequence relative to the domain profile. Annotating the highest-scoring match state at each residue preserves nested domain structure in the ground truth annotations (Fig. 4).

In a match string, regions with majority matches may contain a few inserts and vice versa. To prevent frequently alternating annotations in the ground truth, we smooth the match and insert states in the

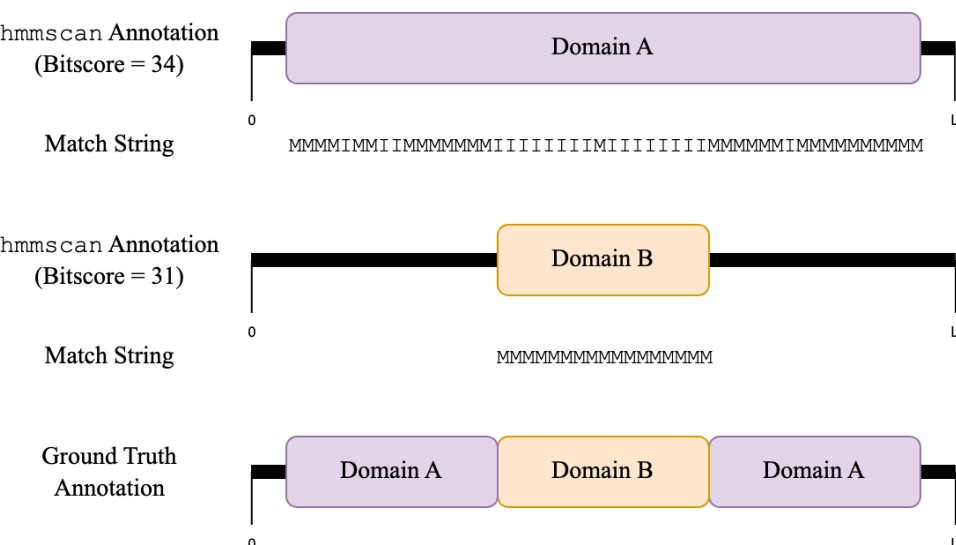

Figure 4: **A schematic of nested domains with two domains A and B in the nested format A-B-A.** As A is annotated with a higher score than B and overlaps with B, annotating residues only via highest score will fail to include domain B. Using the match state strings to identify smoothed maximal segments preserves the nested domain structure in the ground truth annotation.

match string by identifying maximal scoring segments within the sequence. We assign insert states a positive score and match states a negative score. The segment of the sequence with the greatest aggregate score is known as the maximal segment (Karlin & Altschul, 1990), and all residues in the maximal segment are denoted as insert states. The scores $s_i$ for each state are inferred from the match string for a given sequence:

$$s_i \propto \log\left(\frac{q_i}{p_i}\right), \tag{5}$$

where $p_i$ is the frequency with which the state appears in the match string, and $q_i$ is a state's target frequency, with $\sum_i p_i = 1$ and $\sum_i q_i = 1$. We set the insert state target frequency at 0.85, the match state target frequency at 0.15, and the length threshold at 20, below which maximal segments are ignored.

### A.1.3 MAXIMUM PID CALCULATION

Once the full length test sequences have been retrieved and subsequently filtered, we compute, for each test sequence, the maximum PID between any of its annotated domains and any annotated domain in train via Algorithm 1. Each test sequence is assigned to a single (out of five) test subset based on its maximum PID.

---

**Algorithm 1** Percent identity splitting test set

---

**Require:** train sequences $\mathcal{D}^{tr}$, test sequences $\mathcal{D}^{te}$, Pfam family profile HMMs $\mathcal{F}$
  Initialize an empty dictionary-like structure record_max_pids
  **for** $f \in \mathcal{F}$ **do**
    train_domains $\leftarrow$ `hmmsearch` $f$ against $\mathcal{D}^{tr}$
    test_domains $\leftarrow$ `hmmsearch` $f$ against $\mathcal{D}^{te}$
    **for** (domain,sequence_id) $\in$ test_domains **do**
      MSA $\leftarrow$ `hmmalign` domain to train_domains with $f$
      domain_pids $\leftarrow$ `esl-alipid` MSA
      max_pid $\leftarrow$ max(domain_pids)
      **if** sequence_id not in record_max_pids **then**
        record_max_pids[sequence_id] $\leftarrow$ max_pid
      **else if** max_pid > record_max_pids[sequence_id] **then**
        record_max_pids[sequence_id] $\leftarrow$ max_pid
      **end if**
    **end for**
  **end for**
  Assign each sequence in $\mathcal{D}^{te}$ to a test split based on max pid

---

The `esl-alipid` tool calculates PID for all pairs of sequences for a given MSA, and is part of the EASEL software package, which can be downloaded together with HMMER (Eddy, 2011).

### A.2 FAMILY-ONLY PSALM FULL RESULTS

We study the effects of predicting clan-level annotations in PSALM by training the PSALM models (across all tested ESM-2 model sizes) without any intermediate predicted clan annotations (Table 5). A subset of this table was included at a fixed FPR of 0.01 in Table 4. Without the intermediate clan predictions, PSALM_$F_{650}$ is only competitive with HMMER* at the clan level for the 0-20% max PID range test subset, though PSALM_$F_{650}$ achieves the lowest max FPR across all test subsets.

Table 5: Residue-level domain annotation benchmark on Pfam Seed dataset

| PID | Model | Clan | | | | Family | | | |
|-----|-------|------|------|------|------|--------|------|------|------|
| | | TPR | FPR | F1 | MCC | TPR | FPR | F1 | MCC |
| 0-20% | HMMER* | 0.694 | 0.033 | **0.819** | 0.642 | **0.659** | 0.033 | **0.810** | 0.636 |
| | PSALM_$F_{650}$ | **0.701** | **0.015** | 0.827 | **0.664** | 0.632 | **0.015** | 0.811 | **0.653** |
| | PSALM_$F_{150}$ | 0.630 | 0.034 | 0.781 | 0.596 | 0.540 | 0.034 | 0.753 | 0.576 |
| | PSALM_$F_{35}$ | 0.560 | 0.065 | 0.733 | 0.523 | 0.412 | 0.065 | 0.670 | 0.476 |
| | PSALM_$F_8$ | 0.394 | 0.115 | 0.599 | 0.355 | 0.232 | 0.115 | 0.469 | 0.253 |
| 20-40% | HMMER* | **0.907** | 0.043 | **0.941** | **0.862** | **0.876** | 0.043 | **0.939** | **0.861** |
| | PSALM_$F_{650}$ | 0.781 | **0.011** | 0.878 | 0.764 | 0.747 | **0.011** | 0.873 | 0.760 |
| | PSALM_$F_{150}$ | 0.705 | 0.032 | 0.833 | 0.691 | 0.651 | 0.032 | 0.822 | 0.682 |
| | PSALM_$F_{35}$ | 0.662 | 0.058 | 0.800 | 0.632 | 0.581 | 0.058 | 0.778 | 0.614 |
| | PSALM_$F_8$ | 0.479 | 0.102 | 0.674 | 0.462 | 0.356 | 0.102 | 0.605 | 0.406 |
| 40-60% | HMMER* | **0.951** | 0.058 | **0.957** | **0.898** | **0.921** | 0.058 | **0.956** | **0.896** |
| | PSALM_$F_{650}$ | 0.833 | **0.012** | 0.906 | 0.810 | 0.816 | **0.012** | 0.904 | 0.809 |
| | PSALM_$F_{150}$ | 0.785 | 0.025 | 0.877 | 0.759 | 0.758 | 0.025 | 0.873 | 0.756 |
| | PSALM_$F_{35}$ | 0.746 | 0.048 | 0.846 | 0.703 | 0.708 | 0.048 | 0.839 | 0.697 |
| | PSALM_$F_8$ | 0.594 | 0.087 | 0.749 | 0.557 | 0.520 | 0.087 | 0.723 | 0.535 |
| 60-80% | HMMER* | **0.974** | 0.059 | **0.971** | **0.924** | **0.946** | 0.059 | **0.970** | **0.923** |
| | PSALM_$F_{650}$ | 0.888 | **0.012** | 0.938 | 0.857 | 0.876 | **0.012** | 0.937 | 0.856 |
| | PSALM_$F_{150}$ | 0.842 | 0.025 | 0.910 | 0.801 | 0.823 | 0.025 | 0.908 | 0.799 |
| | PSALM_$F_{35}$ | 0.792 | 0.048 | 0.876 | 0.733 | 0.770 | 0.048 | 0.872 | 0.730 |
| | PSALM_$F_8$ | 0.632 | 0.093 | 0.776 | 0.569 | 0.586 | 0.093 | 0.762 | 0.558 |
| 80-100% | HMMER* | **0.977** | 0.051 | **0.972** | **0.935** | **0.950** | 0.051 | **0.971** | **0.934** |
| | PSALM_$F_{650}$ | 0.892 | **0.010** | 0.940 | 0.875 | 0.887 | **0.010** | 0.939 | 0.875 |
| | PSALM_$F_{150}$ | 0.835 | 0.024 | 0.903 | 0.808 | 0.819 | 0.024 | 0.901 | 0.806 |
| | PSALM_$F_{35}$ | 0.740 | 0.047 | 0.839 | 0.701 | 0.720 | 0.047 | 0.835 | 0.697 |
| | PSALM_$F_8$ | 0.661 | 0.078 | 0.783 | 0.609 | 0.627 | 0.078 | 0.774 | 0.601 |

