# OpenReview forum: "Protein Sequence Domain Annotation using Language Models"
_ICLR.cc/2025/Conference — ICLR 2025 Conference Withdrawn Submission_

### Official Review · Reviewer_WVA9 · 2024-10-29

**Soundness:** 2
**Presentation:** 3
**Contribution:** 2
**Rating:** 3
**Confidence:** 4

**Summary:**

The described study includes the training of a residue-level domain prediction model, PSALM, built on the pre-trained ESM-2 pLM. In addition, the authors provide a benchmark dataset resource for future evaluation of other models in this task, MDPH-Bench. The model itself leverages the hierarchical classification of PFAM domains which are characterized at the clan and family level. The PSALM architecture consists of two prediction tasks per protein residue: 1) the probability of belonging to a clan and 2) the probability of a belonging to a family. The MDPH-Bench is a curated set of domains from PFAM that restricts the pairwise sequence percent identity to <25% with some domain and sequence exclusion criteria.

The main contribution of the work is a residue level domain scanner that provides probability over a set of 560K domains. It can predict multiple domains in a sequence, like HMM-based methods, and unlike prior work described. The application provides a useful tool for researchers that allows for leveraging a pre-trained pLM for a sequence-based search. The experiments demonstrate that at low PID, PSALM can identify domains that current approaches cannot, and the authors provide two examples of sequences annotated by their approach and potential advantages compared to an HMM-based method.

**Strengths:**

The paper is generally clear in presentation and easy to follow. It addresses a clear need in the community and provides an additional resource in the form of a benchmark dataset for the task.

**Weaknesses:**

While the application is described as novel, the utilization of pre-trained pLMs for transfer learning of sequence-based tasks and annotation is not novel and modeling approach and architecture are not novel either as cited in the manuscript.

Of most concern, is the inclusion of clan level information in training makes this modeling approach dependent on alignment based annotation and limits the ultimate comparison to HMM-based ground truth annotation of family. The assignment of annotation by profile-HMM is a statistical comparison and dependent on the size of the database searched (e-value) with a pre-defined threshold, in this case < 0.001. When profile-HMM libraries are "trained" they are not provided clan level information but in the case of PSALM that information is provided to the model at training. It would be more accurate to compare PSALM against a database of clans and then assign ground truth families using a profile-HMM search against families in that clan. The authors attempt to address this point with the experiment where they trained PSALM_F without clan level information, however the results here do no show superior performance for the two-task PSALM base architecture (see Table 5) and PID 0-40% for fixed FPR of 0.01 (Table 4). Further discussion of the inclusion of clan in training the family model is warranted.

Finally, there is no evaluation of the model outside of the per-residue performance, which diminishes the significance of this work. In Section 6, it is unclear how these two sequences were chosen for evaluation. Across the whole test set, how many domains were predicted where InterPro did not predict them? Additionally, it would be useful to see what domains the model is recovering that HMMs do not. Also, what position are the residues that the model does recover but the HMMs do not?

**Questions:**

1) In Figure 2, why does it appear that the true clan vector (z) is being passed into the second LSTM, based on Equation (3) is it not z_hat that is passed to the second LSTM?
2) I struggled to understand how the MDPH-Bench was constructed. It would be useful to have a flow diagram that explains the inclusion/exclusion criteria for sequences and then the division into PID sets.
3) How was the PID of 25% decided? It would be good to see some evidence for why this is the threshold. Maybe the authors could try some text based method to show that the overlap in domain annotation decreases with PID threshold.
4) Can you justify why the FPR is fixed at 0.01 for the experiment with results reported in Table 4? It looks like HMMER achieves optimal F1 with FPR in the 0.03-0.06 range (Table 3) and that HMMER* does have superior performance in that range (Table 5).
5) Figure 3B shows some considerable mixed annotation of the region between the two N terminal domains, is this a common phenomenon?
6) By selecting the clan and domain using argmax(), it allows for classification when probabilities are low if there are multiple classes with density, have the authors thought about this? For example, in Figure 3B there are some residues with probability < 0.5 for the assignment.

Minor comments:
-duplicate language in the introduction (ll.42-48)
-the acronym PID is not explicitly defined (l.237)

---

### Official Review · Reviewer_uAsi · 2024-11-03

**Soundness:** 4
**Presentation:** 3
**Contribution:** 3
**Rating:** 5
**Confidence:** 4

**Summary:**

The paper describes PSALM (Protein Sequence Annotation using Language Models), a hierarchical approach that is an alternative approach to profile HMMs (pHMM), the current state-of-the-art for protein domain-based homology detection. PSALM uses representations of protein sequences learned by protein language models to do residue-level protein sequence annotation, a common and important task for biologists. The authors also develop a benchmarking dataset for remote, multi-domain,  protein homology detection tasks (MDPH-Bench). Other approaches, both sequence alignment based (pHMM) and convolutional neural network models (ProtENN) identify one domain at a time for a particular sequence, whereas PSALM can predict multiple domains in a sequence. Performance of PSALM is meaningfully stronger in the 0-20% PID category, which is where the most difficult to identify homologous sequences are found. In the other categories PSLAM and HMMER are doing on par with each other with very small differences. The PSALM annotation examples shown are in the higher PID categories, not in the lower, and the authors note that one example was removed from the Uniprot database, why was this?

**Strengths:**

PSALM uses representations of protein sequences learned by protein language models to do residue-level protein sequence annotation, a common and important task for biologists.

The authors also develop a benchmarking dataset for remote, multi-domain,  protein homology detection tasks (MDPH-Bench). Other approaches, both sequence alignment based (pHMM) and convolutional neural network models (ProtENN) identify one domain at a time for a particular sequence, whereas PSALM can predict multiple domains in a sequence.

Performance of PSALM is meaningfully stronger in the 0-20% PID category, which is where the most difficult to identify homologous sequences are found.

**Weaknesses:**

The novel annotation examples shown are in the higher PID categories, not in the lower where the PSALM does far better than HMMs

The performance on categories higher than 0-20% PID is generally very similar between HMM and PSALM. Given that HMMs are more 'explainable' than PSALM (they are based on multiple sequence alignments), why would one use PSALM on these higher PID classes?

The authors note that one example was removed from the UniProt database, why was this? Why not use an example for a protein sequence that is in this database, which was used for their training?

**Questions:**

Why didn't you show annotation examples from the 0-20% PID group? This class of sequences seems to be where PSALM can annotate things that HMMs can't annotate.

Why do you think PSALM has similar performance for the higher PID groups to the HMMs? Are HMMs identifying all/most of the 'information' needed for annotation? Can you use a combination of PSALM/HMM to do better in these groups?

Why is your example for Uniprot not in the database any longer/why not use another example?

---

### Official Review · Reviewer_yrNk · 2024-11-03

**Soundness:** 3
**Presentation:** 2
**Contribution:** 3
**Rating:** 5
**Confidence:** 4

**Summary:**

This paper presents PSALM (Protein Sequence Annotation using Language Models), a novel approach that leverages embeddings from the pretrained ESM-2 language model to improve protein domain annotation at a residue level. PSALM introduces a hierarchical clan-family prediction structure, which first assigns a broader clan label before refining it to a specific family label for each residue. This setup aims to enhance sensitivity and specificity in detecting protein domains, especially in challenging multi-domain proteins and distantly related homologs.

To evaluate PSALM, the authors created MDPH-Bench, a benchmark designed for testing multi-domain protein annotation with strict percent identity (PID) splits between training and test sets. This benchmark allows for performance evaluation across different levels of evolutionary similarity. The paper compares PSALM to HMMER, a standard HMM-based tool for domain annotation, and demonstrates that PSALM achieves higher sensitivity and specificity, particularly in challenging, low-PID ranges.

**Strengths:**

The paper introduces a novel application of pLMs (ESM-2 embeddings), in a hierarchical clan-family prediction framework for domain annotation, especially for multi-domain and low-similarity sequences.

Additionally, the introduction of MDPH-Bench adds a unique contribution that can be valuable for evaluating future models in this area.
The paper demonstrates technical rigor, with detailed evaluations across different PID ranges to show the model’s strengths in various contexts. The authors also include ablation studies to highlight the specific contribution of ESM-2 embeddings and clan-level learning, adding depth to the analysis.

The work has practical implications for protein domain annotation, an important task in bioinformatics. By addressing limitations of HMM-based methods, PSALM and MDPH-Bench add tools that could support both research and applied work in areas like functional genomics and evolutionary studies.

**Weaknesses:**

I have mixed feelings about this paper, as it introduces several interesting ideas. However, due to issues with clarity, not being fully self-contained, and its potential lack of fit for this venue, I am inclined to not give it an "accept" score. If these comments are addressed, I would be willing to raise my score.

The paper is aimed at a bioinformatics audience with substantial familiarity with domain annotation and protein language models. For ICLR, a machine learning-focused conference with a broad readership, the presentation lacks accessibility.
The paper needs to be self-contained, providing clear explanations of key terms and methods to ensure that a general ML audience can understand it without additional background knowledge. Foundational concepts, such as HMMER’s “simplifying assumptions,” are not explained in enough detail, sometimes missing entirely, leaving gaps that could hinder comprehension. HMMER is used both as a baseline and ground truth annotation and this could lead to confusion, especially for readers outside of bioinformatics. This distinction should be made clearer. Even for bioinformatics, this is a narrow area, and it should be properly explained.
While PSALM shows a creative application of ESM-2 embeddings for domain annotation, the paper does not contribute new methods or findings to representation learning itself, which is central to ICLR’s focus.

Additionally, the paper’s high computational requirements are not addressed with any comparisons in terms of runtime or resource efficiency, which would be essential to assess its practicality for large-scale applications. Given PSALM's reliance on ESM-2 embeddings and BiLSTM layers, understanding how its performance gains weigh against the increased computational cost would make the model’s contributions clearer for readers considering its application in real-world settings.

**Other comments:**

- Some figures and tables, especially Figure 2 which is the main figure explaining the method, are not referenced in the main text. Figure 2 can be used in the method explanation to more accurately lead the reader in understanding the method.
- There is a repeated sentence in the 2nd paragraph of the introduction.
- In the introduction, the authors cite some references when discussing protein function prediction. While these are valid references, they seem outdated considering the recent advancements in this field.
- In the results section, the authors mention that "biologists prefer domain-level annotation for many reasons." While this may be accurate, statements like this should be properly referenced, especially given the interdisciplinary audience at this venue.

**Questions:**

•	The use of a BiLSTM for PSALM’s architecture feels somewhat under-motivated. The authors mention that this choice was made deliberately to introduce as few changes as possible from profile HMMs, allowing observed improvements to be attributed primarily to the PLM rather than architectural differences. However, HMMs are inherently linear models, whereas BiLSTMs are nonlinear, which could introduce significant architectural differences that may influence the results. Additionally, given that ESM-2 embeddings are already position-aware, wouldn’t a simpler architecture, such as a feed-forward neural network for clan and family prediction, offer a more direct evaluation of the embeddings' impact? Could the authors clarify the rationale for choosing a BiLSTM over a simpler model and discuss any experiments or considerations around this choice?

•	The authors discuss "scalability" by evaluating performance across different ESM-2 model sizes, increasing the architecture's number of trainable parameters. However, scalability typically refers to a model's ability to handle larger datasets or tasks efficiently, without a proportional increase in computational resources. In this case, expanding model size enhances model capacity rather than scalability, as it increases computational demand rather than demonstrating efficient scaling.
Could the authors clarify this terminology and, if appropriate, provide resource requirements and prediction times for each model size on different test set sizes? This would help evaluate PSALM’s practicality in real-world applications and give a clearer sense of its performance trade-offs across model configurations.

•	In Table 4, HMMER shows an unusual performance pattern, performing worse in high-PID test sets and achieving its best results in remote homolog test sets (lower PID). This is counterintuitive, as one would typically expect HMMER to perform better with higher PID values, given that profile HMMs are generally more effective on closely related sequences. Could the authors elaborate on this outcome and provide intuition or hypotheses as to why HMMER might exhibit better performance on lower PID test sets?

---

### Official Review · Reviewer_W8wY · 2024-11-04

**Soundness:** 3
**Presentation:** 3
**Contribution:** 3
**Rating:** 5
**Confidence:** 4

**Summary:**

This study introduces PSALM (Protein Sequence Annotation using Language Models) for protein domain annotation and develops the Multi-Domain Protein Homology Benchmark (MDPH-Bench) as a benchmark for this task. The proposed method was validated on MDPH-Bench, and the authors present PSALM as a promising alternative to HMMER for protein sequence annotation.

**Strengths:**

The layout is clear, and the presentation is well-organized.

**Weaknesses:**

1. It appears that the authors may not be fully acquainted with the protein domain annotation task. The paper would benefit from a more comprehensive literature review and a discussion of additional baseline methods.
2. Protein domain annotation is crucial for understanding protein function. In addition to HMMER, there are many existing methods, such as LSTM-based and structure-based approaches. The authors should review these methods and provide a comparative analysis.
3. Since protein domain annotation is a classical task, the detailed problem formulation may not be necessary.

**Questions:**

As shown in the weakness.

---

### Note · Authors · 2024-11-26

**Comment:**

We thank the reviewers and chairs for their time and their thoughtful comments and suggestions on how best to improve our work. In light of these reviews, we have decided to withdraw this manuscript from ICLR.

**Withdrawal Confirmation:**

I have read and agree with the venue's withdrawal policy on behalf of myself and my co-authors.